# Advanced Imaging in Cardiac Amyloidosis

**DOI:** 10.3390/biomedicines10040903

**Published:** 2022-04-15

**Authors:** Dominik Waldmeier, Jan Herzberg, Frank-Peter Stephan, Marcus Seemann, Nisha Arenja

**Affiliations:** 1Kardiologie Solothurner Spitäler, 4500 Solothurn, Switzerland; dominik.waldmeier@bluewin.ch (D.W.); jan.herzberg@spital.so.ch (J.H.); frank-peter.stephan@spital.so.ch (F.-P.S.); 2Radiologie Bürgerspital Solothurn, 4500 Solothurn, Switzerland; marcus.seemann@spital.so.ch

**Keywords:** cardiac amyloidosis, echocardiography, CMR, cardiac scintigraphy, PET

## Abstract

This review serves as a synopsis of multimodality imaging in cardiac amyloidosis (CA), which is a disease characterized by deposition of misfolded protein fragments in the heart. It emphasizes and summarizes the diagnostic possibilities and their prognostic values. In general, echocardiography is the first diagnostic tool in patients with an identified systemic disease or unclear left ventricular hypertrophy. Several echocardiographic parameters will raise suspicion and lead to further testing. Cardiac magnetic resonance and scintigraphy with bone avid radiotracers are crucial for diagnosis of CA and even enable a distinction between different subtypes. The subject is illuminated with established guidelines and innovative recent publications to further improve early diagnosis of cardiac amyloidosis in light of current treatment options.

## 1. Introduction

Cardiac amyloidosis (CA) is a disease characterized by deposition of misfolded protein fragments—called amyloid fibrils—in the heart. The two most frequent types of amyloidosis with cardiac involvement are light-chain amyloidosis (AL) and transthyretin amyloidosis (ATTR) [1]. TTR amyloidosis incorporates two subtypes: ATTRv (variant or hereditary amyloidosis) stands for all hereditary forms, whereas ATTRwt (wild-type amyloidosis) is defined by sporadic mutations of the transthyretin gene [2]. Secondary amyloidosis is a rare type of amyloidosis with deposition of serum amyloid A. It generally appears along with inflammatory diseases such as rheumatoid arthritis [3].

Although AL and ATTR have similar effects on myocardial function and patients typically present with signs and symptoms of heart failure, they are different conditions with various therapeutic approaches. Thereby, early initiation of therapy is crucial for the prognosis [4]. This is particularly worthwhile given the recent significant developments of treatment options for both types of amyloidosis.

Endomyocardial biopsy is still considered the diagnostic ‘gold standard’, leading to the histological diagnosis of CA by Congo red staining. Despite its invasive nature, it is considered a fairly safe procedure if performed by an experienced examiner [5,6]. Due to the remaining risk, noninvasive techniques for diagnosis are preferably used. A lot of research has been conducted towards finding reliable markers for early diagnosis of CA using advanced imaging.

The purpose of this review is to summarize recent developments in available imaging modalities, highlight their advantages and disadvantages, provide a statement of diagnostic and prognostic value, and offer insight into recent work of various researchers. Table 1 offers an overview of the above.

## 2. Echocardiography

### 2.1. Transthoracic Echocardiography

Transthoracic echocardiography (TTE) is widely available and can be performed in any setting, making it an excellent screening tool for initial evaluation. Increased thickening of the left ventricular wall, the interatrial septum, as well as the valves, and the presence of pericardial effusion are typical findings suggestive of CA (Figure 1). In addition, the assessment of diastolic function, which is usually impaired (Grade 2 or more), is of great importance and only feasible with TTE [7].

Michele Boldrini and colleagues provide a multiparametric TTE approach for differentiation of patients with systemic AL from a hypertrophic phenotype. They found specific functional and structural TTE parameters characterizing different burdens of CA deposition to diagnose or exclude the diagnosis. The algorithm was developed to avoid unnecessary tests and limit the time to diagnosis, which is crucial for the patient’s prognosis [8].

Strain evaluated by speckle tracking echocardiography (STE) presents an additional feature in the diagnosis of CA. A ‘speckle’ of myocardium identified by software is followed through a cardiac cycle to reveal information on its radial, longitudinal, and circumferential deformation. Therefore, a certain degree of image quality is necessary for conduction of STE. The characteristic STE pattern shows a reduction in global longitudinal strain (GLS) of the left ventricle in the basal and mid-ventricular segments with an apical sparing also called the ‘cherry-on-top’ sign and preserved left ventricular ejection fraction (LVEF).

A small Spanish study of 40 patients with CA examined the performance of STE compared to cardiac biomarkers, such as brain natriuretic peptide (BNP) and cardiac troponin I, in the early diagnosis of CA. Strain values of both the right and the left ventricle seem to perform better than cardiac biomarkers [9]. Pagourelias et al. studied 100 patients (40 with confirmed CA, 40 with hypertrophic cardiomyopathy (HCM), and 20 with hypertensive heart disease) in order to assess the diagnostic value of different TTE parameters. Deformation parameters such as global longitudinal/radial/circumferential strain, relative apical sparing, and septal apical to base longitudinal strain have shown better results in differentiating between patients with CA and hypertrophic hearts of other causes compared to traditional parameters. Ejection fraction to strain ratio (EFSR) and myocardial contraction fraction (MCF, ratio of stroke volume and myocardial volume) achieved the best results for deformation parameters and traditional parameters, respectively, also in patients with otherwise normal LVEF and mild LV wall thickness [10].

Despite the aforementioned possibilities, including the use of algorithms with highly sensitive and specific parameters, in most cases it is hardly possible to diagnose CA with echocardiography alone and further evaluation is needed.

### 2.2. Transesophageal Echocardiography

Transesophageal echocardiography can be performed for diagnosis in patients with suspected thrombus formation, but has an otherwise subordinate role compared to TTE. The formation of intracardiac thrombi, especially in the left atrium and the left atrial appendage, occurs frequently in patients with CA. Feng et al. examined 116 patients with CA post-mortem and found intracardiac thrombi in 33% of cases [11]. Reduced blood flow velocity in the left atrial appendage explains this phenomenon. This is due to impaired diastolic function of the left ventricle, impaired mechanical function of the dilated left atrium, but also atrial fibrillation [12].

## 3. Cardiovascular Magnetic Resonance

Cardiovascular magnetic resonance (CMR) has several advantages compared to TTE. In most centers, it is used for further investigation of subjects with suspected CA and in patients with confirmed systemic amyloidosis. On one hand, it offers unmatched precision when it comes to measurements, including the assessment of the right ventricle; on the other hand, there is a unique opportunity of tissue characterization. The interobserver variability is presumably low, even in patients with a poor acoustic window. However, this does not apply to patients with cardiac devices or arrhythmias, which may impair the image quality due to artifacts.

Standardized protocols for image acquisition are strongly recommended by the Society of Cardiovascular Magnetic Resonance (SCMR) [13]. They normally include cine images, native and post-contrast T1 mapping, T2 mapping, and the administration of a gadolinium contrast agent using steady-state free precession sequences (SSFP) (Figure 2). Indications for CA arise from the abnormalities in the standard measurements described in detail below.

### 3.1. Chamber Quantification

Compared to TTE, CMR provides more precise information about structural and functional abnormalities. Cine images are acquired by merging a number of stationary images taken at different cardiac cycles. They are reconstructed to map views familiar from TTE studies (e.g., two-, three-, and four-chamber views or short-axis views). The recommended measurements do not greatly differ from TTE assessment and include wall thickness, function and mass of the left ventricle, stroke volume index to body-surface-area, atrial size, and the presence or absence of pericardial effusion [7]. Typical findings in ATTR CA are higher LV mass index, higher LV volumes, and lower LVEF [14,15].

In addition to standard measurements, there is a trend towards process optimization using complex computer software for fully automated measurements, often based on machine learning algorithms. However, these have not yet reached the routine clinical process, so conventional analysis methods are still required. Our research group tried to find easy parameters derivate from CMR images independent of the aforementioned post-processing software. We evaluated the prognostic value of long-axis strain (LAS, image-based functional marker describing the percentage of longitudinal shortening of the left ventricle during systole) and previously mentioned myocardial contraction fraction (MCF) in 74 patients with AL. All-cause mortality (primary endpoint) was significantly higher in patients with low MCF. Both impaired LAS and MCF were preserved as predictors for all-cause mortality and heart transplant [16].

### 3.2. Strain Imaging

In analogy to speckle tracking in TTE, CMR uses feature tracking for strain analysis. ‘Feature’ refers to a certain myocardial pattern recognized and followed through a cardiac cycle by post-processing software to observe radial, circumferential, and longitudinal deformation of the myocardium. Like in STE, patients with CA being evaluated with CMR feature tracking typically show a pattern of relative apical sparing.

In a study of 83 patients, Williams et al. compared CMR strain in patients with CA (45 patients), HCM, and Fabry disease (19 patients each for HCM and Fabry disease, respectively). According to their findings, CA is associated with significantly lower longitudinal strain compared to both diseases. They also assessed the relative regional longitudinal strain ratio (RRSR, average apical longitudinal strain/(average basal longitudinal + average mid longitudinal strain)) as an alternative parameter of the basal to apical strain gradient, with significant differences between CA and Fabry disease only [17].

A small comparative study of 40 patients (20 individuals with CA and HCM each) conducted research on rotational mechanics of the left ventricle also using CMR feature tracking. The systolic movement around a longitudinal axis of the left ventricle is referred to as left ventricular twist (LVT). During systole, the LV base and the LV apex move in a clockwise and anticlockwise direction, respectively. The authors were able to demonstrate impaired LVT in patients with CA and increased LVT in patients with HCM [18].

### 3.3. T1 Mapping

T1 mapping is an emerging tool used for tissue characterization and includes two sequences: native or pre-contrast T1 and post-contrast T1. Images are recommended to be acquired during diastole [5]. Variations of Look-Locker inversion recovery sequences were introduced to overcome issues with insufficient breath holding in patients. T1 relaxation times vary with magnetic field strength, and myocardial T1 relaxation times also depend on the patient’s age and sex [17]. Relaxation times are color-coded for every pixel and assembled on a two-dimensional plane, creating a T1 map [17].

Patients with systemic amyloidosis often have deposition of amyloid fibrils in the kidneys, leading to relevant renal dysfunction. This is an important limitation for the use of intravenous contrast agents. For the acquisition of native T1, none of them are necessary. It therefore takes an important diagnostic role when contrast agents are contraindicated. Native T1 displays not only extra- but also intra-cellular parts of the myocardium. Elevated T1 times are associated with the presence of CA. A very recent publication studied a large group of patients with suspected systemic amyloidosis. Though applied to selected individuals, high sensitivity and specificity were achieved (92% and 91%, respectively) [19].

T1 mapping is also believed to be a reliable tool for the monitoring of amyloid burden in AL during chemotherapy [20].

Extracellular volume (ECV) is defined as extracellular matrix, water, and intracapillary plasma volume. ECV, but also intracellular volume, are dynamic and vary for different physiological and pathophysiological processes. Combining pre- and post-contrast T1 mapping enables the calculation of ECV. The patient’s hematocrit needs to be measured for that purpose. A normal ECV value for myocardium is around 25% [17]. The deposition of amyloid fibrils in CA leads to an increase of ECV.

Korthals et al. compared receiver operating curves of native T1 and ECV as well as longitudinal strain and were able to illustrate superior diagnostic performance of native T1 and ECV over longitudinal strain [21]. They earlier emphasized that special attention needs to be drawn towards choosing an adequate control group to define cut-off values [22]. Recommendations by the SCMR and the European Association for Cardiovascular Imaging (EACVI) stated that native T1 and ECV are increased before LGE, enabling earlier diagnosis of CA [23]. A meta-analysis by Pan et al. published in 2020 proved the diagnostic performance of ECV to be better than both native T1 and LGE. The same statement can be translated to the prognostic performance [24].

### 3.4. Late Gadolinium Enhancement

Gadolinium-based contrast agents (GBCA) are administered intravenously following predefined protocols. In the literature, cases of nephrogenic systemic fibrosis, a rare kidney disease with poor prognosis, after the use of GBCA are described. It remains a controversial subject [25]. Recent evidence reports deposition of gadolinium in the brain after administration of GBCA, irrespective of renal function and performance of the blood–brain barrier. There are no reports of neurologic deficits due to gadolinium deposition. As a consequence, monitoring of renal function is mandatory, and the use of cyclic GBCA instead of linear agents is recommended [26]. The high prevalence of renal disease in patients with suspected amyloidosis is an important limitation in the use of GBCA. Standards suggest that images are acquired more than ten minutes after the intravenous administration of the contrast agent. The recommended dose depends on the chosen agent and should be as low as possible considering the aforementioned potential complications [11].

Phase-sensitive inversion recovery (PSIR) is used to overcome problems with ‘myocardial nulling’. Difficulties in ‘myocardial nulling’ are typical for CA and describe the challenge of finding an inversion time to differentiate the myocardium from the blood pool when acquiring images with GBCA [27].

Myocardial uptake of gadolinium is referred to as late gadolinium enhancement (LGE). The accumulation of amyloid fibrils leads to the expansion of extracellular space, where gadolinium uptake experiences regional differences depending on the extent of expansion. Patterns range from typically described subendocardial to diffuse, transmural, or patchy dispersion. Though discrimination between different types of CA is not reliably feasible, a study including almost one hundred patients with cardiac involvement of amyloidosis demonstrated that LGE is more extensive in patients with ATTR and there is a higher proportion of patients showing transmural patterns of LGE [14]. Transmural LGE patterns are suggestive for advanced stages of the disease [28]. To our knowledge, there is no established staging system involving LGE.

### 3.5. T2 Mapping

T2 mapping is another quantitative tool used for tissue characterization. T2 times are tissue-specific and enable differentiation between normal and abnormal myocardial tissue. Water has quite a long T2 time. Therefore, the tissue has a higher content of water, hence myocardial edema can be depicted by this technique. Reference values for T2 times are different for 1.5 and 3 T MRT scanners [29]. Kotecha et al. proved that T2 mapping can be used as an independent predictor of prognosis in patients with cardiac AL. The study also suggests that different types of CA are part of a heterogeneous group of diseases, since there are significant differences between untreated individuals with AL, treated AL, and ATTR [30].

A meta-analysis by Brownrigg et al. analyzed the performance of CMR in differentiating AL and ATTR (sensitivity and specificity of 28.1–99% and 11–60%, respectively) and compared CMR to scintigraphy (sensitivity and specificity of 88.6–90.9% and 91.5–97.1%, respectively) [31]. In summary, CMR is a useful diagnostic tool for evaluation of CA, but it does not currently allow the differentiation between the different types of amyloidosis.

## 4. Nuclear Imaging

Nuclear imaging is a generic term for several imaging techniques. They all make use of small amounts of radioactive tracers administered to gain information about organs and tissues and represent an additional pillar in the diagnosis of CA. Nuclear imaging is not about functional, but more about structural assessment. While scintigraphy is two-dimensional or planar, positron emission tomography (PET) and single-photon emission computed tomography (SPECT) are three-dimensional.

### 4.1. Scintigraphy and Single-Photon Emission Computed Tomography

Scintigraphy is recommended in patients with suspected CA due to findings from TTE and CMR. In addition, patients with confirmed ATTR neuropathy or a positive test for a TTR gene mutation are also advised to undergo scintigraphy. Otherwise, it should also be considered in case of unexplained left ventricular wall thickening, bilateral carpal tunnel syndrome, and for individuals with a family history of amyloidosis [7].

Today, three bone avid radiotracers are used to perform scintigraphy in patients with suspected CA [32]. All of them are based on the radioisotope 99mTechnetium (99mTc). 99mTc-pyrophosphate (PYP) is approved by the Food and Drug Administration (FDA) and mainly used in the United States (US). 99mTc-hydroxymethylene diphosphonate (HMDP) and 99mTc-3,3-diphosphono-1,2-propanodicarobxylicacid (DPD) are applied in Europe, but rarely or not utilized in the US. The radiotracers are injected intravenously. Whole-body planar images and chest or cardiac SPECT images are obtained two to three hours post-injection. The myocardial uptake of the tracer is then compared to the uptake of the rib cage (typically as a heart-to-bone and heart-to-contralateral ratio). It is graded from 0 to 3 according to the Perugini grading scale (0: no uptake, 1: uptake less than rib, 2: uptake equal to rib, 3: uptake greater than rib). Grades 2 and 3 are strongly suggestive for the presence of cardiac ATTR. Grade 0 is not suggestive, and Grade 1 is considered equivocal (Figure 3). Scintigraphy should always be accompanied by the performance of serum and urine immunofixation and serum-free light-chain studies. In September 2019, the American Society of Nuclear Cardiology (ASNC) and the European Association of Nuclear Medicine (EANM) published practice points regarding scintigraphy in CA.

In the late twentieth century, knowledge about scintigraphy for the diagnosis of CA was not consolidated. A study from 1987 by Gertz et al. serves as an example by showing that only a loose association between scintigraphy and CA was established [33]. Further research revealed, both in amyloidosis and scintigraphy, that microcalcifications are responsible for the selectivity of bone avid radiotracers in CA. The density of microcalcifications is higher in ATTR compared to AL [34].

A large study published in 2016 highlighted the performance of scintigraphy for the diagnosis of cardiac ATTR. The combined finding of Grade 2 or 3 radiotracer uptake in the absence of monoclonal protein in serum and urine is associated with a specificity and a positive predictive value for ATTR CA of 100%. Radiotracer uptake alone had a sensitivity of >99% and a specificity of 86% for ATTR CA. False positive cases almost exclusively occurred in patients with AL CA [35]. Especially in patients with advanced amyloid deposition, scintigraphy using bone avid radiotracers is to our knowledge the best available tool for the discrimination of AL and ATTR CA [36]. Scintigraphy therefore plays an important role in the guidelines for the diagnosis of CA of both the American Heart Association and the European Society of Cardiology [37,38].

In addition, radiotracer uptake correlates positively with all-cause mortality and negatively with survival free of cardiac adverse events [7]. Another approach, which is used to evaluate the forecast, is the assess autonomic dysfunction as a non-specific finding in patients with CA resulting in arrhythmias. Here, 123I-metaiodobenzylguanidine (MIBG) is used as a radiotracer, analogous to norepinephrine, to derive the cardiac-to-mediastinum ratio. In combination with the heart rate response to atropine, it can be used as a predictor of prognosis in patients with ATTR CA [36,39].

### 4.2. Positron Emission Tomography

11C-Pittsburgh Compound B, 18F-Florbetapir, and 18F-Florbetaben are amyloid-binding radiotracers. They were not designed for cardiac imaging but, i.e., for imaging of beta-amyloid deposits in the brain in Alzheimer’s disease. PET is a functional nuclear imaging technique using computed tomography (CT) or magnetic resonance tomography (MRT), which is therefore not only eligible for CA but also systemic amyloidosis. Amyloid-binding radiotracers enable a quantification of the global amyloid burden but also of a regional burden, such as the myocardium. The parameters measured are the self-explanatory target-to-background ratio, myocardial standard uptake value (SUV, parameter to measure radioactivity of the studied tissue), and myocardial retention index (derived from SUV in different phases after radiotracer injection). The direct binding of the tracers to the agent causing the disease allows early detection of the disease, even before structural changes appear. Early detection is crucial for effective treatment. Distinction between the different types of CA is not feasible with this technique. Due to possible quantification of deposits, PET has the potential for serial imaging in order to monitor response to treatment [40].

Singh and Dorbala provide an overview of several studies conducted on PET in CA. The studies are limited by small sample sizes and a lack of standardized protocols for image acquisition. This can be illustrated with one specific publication commented on in the above work, which studied a new radiotracer (amyloid-binding 18F-Flutemetamol) in only 17 patients undergoing PET (with CT or MRT). Images were acquired more than one hour after injection of the radiotracer, which is rather late compared to other amyloid-binding radiotracers. Only a small percentage of patients with CA showed relevant radiotracer uptake [41,42,43].

However, further research with larger sample sizes and standardized protocols is needed before PET is applied in clinical routine.

## 5. Conclusions

If the suspicion for CA is raised by clinical, laboratory, electrocardiographic, and very often echocardiographic findings, a multitude of imaging techniques can be applied to confirm the diagnosis and to differentiate between different types of amyloidosis non-invasively, such as CMR, scintigraphy, and PET.

CMR is able to reveal the structure and function of the chambers and to quantify amyloid deposits. The administration of GBCA is limited due to the high incidence of impaired renal function in patients with amyloidosis. Scintigraphy with bone avid radiotracers in combination with laboratory testing offers accurate diagnosis of cardiac involvement in ATTR amyloidosis. PET using amyloid-binding radiotracers has a subordinate role but enables early detection and quantification. Advantages and disadvantages of all imaging modalities have been listed in Table 1.

Current guidelines recommend CMR especially for the differentiation of CA and other cardiac pathologies. Scintigraphy should be performed in patients with suspected CA in the absence of a monoclonal light chain. The use of PET is only mentioned in the context of disease progression and the response to treatment [37,38].

CMR and scintigraphy allow an estimation of prognosis. Radiotracer uptake in scintigraphy correlates with all-cause mortality [7]. LGE, native T1, and ECV offer assessment of prognosis in CMR, where the latter is believed to perform the best [25].

## Figures and Tables

**Figure 1 biomedicines-10-00903-f001:**
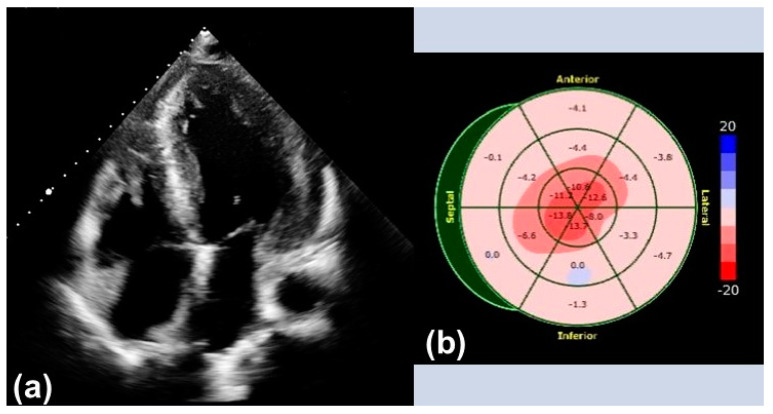
Typical findings in transthoracic echocardiography: (**a**) Apical four-chamber view with left ventricular hypertrophy. (**b**) Echocardiographic strain imaging with ‘cherry-on-top’ sign.

**Figure 2 biomedicines-10-00903-f002:**
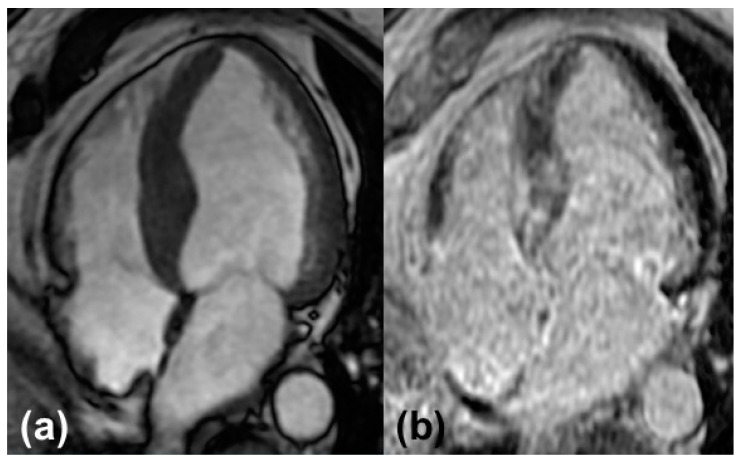
Typical findings in cardiovascular magnetic resonance: (**a**) Cine image of a four-chamber view with left ventricular hypertrophy. (**b**) Late gadolinium enhancement of the left and right ventricular myocardium.

**Figure 3 biomedicines-10-00903-f003:**
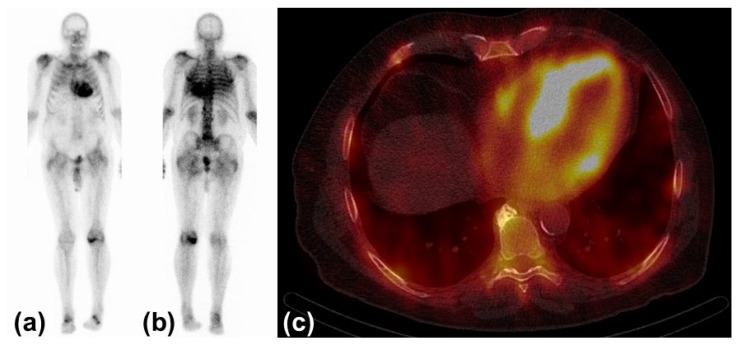
Typical findings in scintigraphy and single-photon emission computed tomography: (**a**) Anterior view of 99Tc-hydroxymethylene diphosphonate scintigraphy with myocardial uptake Perugini Grade 3. (**b**) Posterior view of the same study. (**c**) Left and right ventricular uptake of 99mTc-hydroxymethylene diphosphonate in single-photon emission computed tomography.

**Table 1 biomedicines-10-00903-t001:** Summary of imaging modalities and their features.

	TTE	TEE	CMR	Scintigraphy/SPECT	PET
**Availability**	+++	+++	++	++	+
**Approximate cost**	$	$	$$	$$	$$
**Radiation**	No	No	No	Yes	Yes
**Differentiation between subtypes of CA**	No	No	No	Yes, in absence of monoclonal protein in serum and urine	No
**Information on other causes of LVH**	Yes	No	Yes	No	No
**Role for diagnosis**	Screening and follow-up tool, structural and functional assessment	Only useful for detection of LAA thrombi	Structural and functional assessment, tissue characterization	Valuable for suspected ATTR CA	Not widely used due to limited data
**Typical findings**	Thickening of LV walls, interatrial septum, and valves, diastolic dysfunction pericardial effusion, impaired GLS with apical sparing	Atrial thrombi, even in sinus rhythm	Elevated T1 times, increased ECV, LGE, T2 mapping	Myocardial radiotracer uptake grade 2 or 3	Myocardial radiotracer uptake
**Assessment of prognosis**	No	No	Yes, ECV better than native T1 and LGE, T2 mapping as independent marker for prognosis	Yes, correlation of radiotracer uptake and all-cause mortality	No

TTE = transthoracic echocardiography; TEE = transesophageal echocardiography; CMR = cardiovascular magnetic resonance; SPECT = single-photon emission computed tomography; PET = positron emission tomography; CA = cardiac amyloidosis; LVH = left ventricular hypertrophy; LAA = left atrial appendage; ATTR = transthyretin amyloidosis, LV = left ventricular; GLS = global longitudinal strain; ECV = extracellular volume; LGE = late gadolinium enhancement. Availability: +: limited availability, ++: medium availability, +++: wide availability. Approximate cost: $: low, $$: medium.

## Data Availability

Not applicable.

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
