# Peer review of "Advanced Imaging in Cardiac Amyloidosis"

_biomedicines, 2022, doi:10.3390/biomedicines10040903_

Round 1

Reviewer 1 Report

The authors reviewed articles regarding cardiac amyloidosis by focusing on recent developments in imaging modalities.

This is an interesting review article comprehensively summarizing recent developments in imaging techniques to evaluate cardiac amyloid deposition. Taking up the topic of cardiac amyloidosis is timely because novel disease-modifying therapies for this disease, such as therapeutic agents against plasma cell dyscrasia, transthyretin stabilizers, small interfering RNA, and antisense oligonucleotide, now appear one after another. I do not have any critical comments.

Minor issues and suggestions to strengthen this manuscript are raised as follows: 

  1. Although AL and ATTR amyloidoses are the two major causes of cardiac amyloidosis, the presence of other causes, particularly AA amyloidosis, should be briefly mentioned.
  2. ATTR amyloidosis is comprised of wild-type ATTR (ATTRwt, wt for wild-type) amyloidosis and hereditary ATTR (ATTRv, v for variant) amyloidosis (Cardiol Ther 2021; 10: 289-311). This issue should be clarified in the introduction section by citing relevant articles.
  3. Abbreviations used in a table should be explained in its footnote.

Reviewer 2 Report

The presented manuscript deals with the use of imaging methods in the diagnosis of cardiac amyloidosis. This is a very current issue, which is addressed by many scientists, as evidenced by the number of published articles available, for example, on PubMed.
The manuscript clearly summarizes information about all imaging techniques used in the diagnosis of various types of cardiac amyloidosis published in recent years. The basic characteristics of the use of individual imaging techniques including TTE, TEE, CMR, Scintigraphy/SPECT and PET are summarized in a clear table. I have only a few small points how manuscript could be improved: 1. In chapter 4.1. is information about radiotracers used in United States for the visualization but there is not any information whether in Europe or other parts of the world are used similar or diferent radiotracers. 2. Perhaps some of the images obtained during the examinations with the above methods could bring the reader closer to the described issue.
